# Pregnant women's knowledge of obstetrical danger signs: A cross-sectional survey in Kigali, Rwanda

**Emmanuel Uwiringiyimana**[1]*, **Emery Manirambona**[1], **Samuel Byiringiro**[2], **Albert Nsanzimana**[1], **Neophyte Uhawenayo**[1], **Pacifique Ufitinema**[1], **Janviere Bayizere**[1], **Patricia J. Moreland**[3], **Pamela Meharry**[4], **Diomede Ntasumbumuyange**[5]

1 College of Medicine and Health Sciences, University of Rwanda, Kigali, Rwanda, 2 School of Nursing, John Hopkins University, Baltimore, Maryland, United States of America, 3 Nell Hodgson Woodruff School of Nursing, Emory University, Atlanta, Georgia, United States of America, 4 Human Resource for Health, University of Illinois, Chicago, Illinois, United States of America, 5 Department of Obstetrics and Gynecology, University of Rwanda and Centre Hospitalier Universitaire de Kigali, Kigali, Rwanda

* emma6as@gmail.com

**Data Availability Statement:** All relevant data are within the paper and its Supporting Information files.

## Abstract

Maternal mortality remains critically high in low- and middle-income countries (LMIC), particularly in sub-Saharan Africa. Rwanda's leading causes of maternal death include postpartum hemorrhage and obstructed labor. Maternal recognition of obstetrical danger signs is critical for timely access to emergency care to reduce maternal mortality.To assess maternal knowledge of obstetrical danger signs among pregnant women attending antenatal care services in Kigali, Rwanda. We conducted a cross-sectional study between September and December 2018. The outcome of interest was maternal knowledge of ODS during pregnancy, labor and delivery, and the immediate postpartum period. We recruited pregnant women at five health centers, one district hospital, and one referral hospital, and we had them complete a structured questionnaire. Reporting three correct ODS was defined as having good knowledge of ODS. A total of 382 pregnant women responded to the survey. Most women (48.9%) were aged 26–35, and 50.5% had completed secondary or higher education. The knowledge of ODS was 56%, 9%, and 17% during pregnancy, labor and delivery, and postpartum, respectively. Women aged 26 to 35 had two times (OR: 1.80, 95% CI: 1.05, 3.06) higher odds of ODS knowledge during pregnancy than women aged 16 to 25. Attending three antenatal care visits was associated with 2.6 times (OR: 2.59, 95% CI: 1.17, 5.66) higher odds of ODS knowledge during pregnancy than not attending any visit. Longer distances to the nearby health facility were associated with significantly lower knowledge during pregnancy, and Muslim women had substantially higher postpartum ODS knowledge than any other religion. In conclusion, women's knowledge of ODS associated with labor and delivery and postpartum was low. Antenatal care must be encouraged and its content revised to ensure it covers potential late pregnancy complications.

**Funding:** The authors received no specific funding for this work.

**Competing interests:** The authors have declared that no competing interests exist.

**Abbreviations:** ANC, Antenatal Care; ODS, Obstetrics Danger Signs; DH, District Hospital; HC, Health Center; HEW, Health Expert Worker; MMR, Maternal Mortality Rate; MDG, Millennium Development Goals; PI, Principal Investigator; RDHS, Rwanda Demographic and Health Survey; SPSS, Statistical Package for the Social Sciences; UTHK, University Teaching Hospital of Kigali; WHO, World Health Organization; CI, Confidence Interval.

## Introduction

The complications from pregnancy and childbirth are the leading causes of maternal deaths in low- and middle-income countries (LMIC) [1–3]. Half of all maternal deaths result from post-partum hemorrhage (27%), hypertensive disorders (14%), and sepsis (11%) [1]. Evidence shows that most maternal complications during pregnancy are preventable via timely recognition of danger signs and effective interventions at health facilities [4]. The danger signs during pregnancy and after delivery are the warnings of imminent or ongoing life-threatening events requiring quick intervention by skilled healthcare providers [5]. The lack of knowledge of Obstetrical Danger Signs (ODS) coupled with additional factors such as high cost of care and long distance to health facilities lead to hesitation and delay in seeking care, contributing to higher maternal mortality [6–9]. Any efforts to reduce maternal mortality must include improving maternal knowledge of the common complications during pregnancy, labor, and after delivery to ensure women prioritize seeking care when required.

Several factors predict women's knowledge of ODS and could guide where to target interventions. According to a study conducted in Ethiopia, the attendance of Antenatal Care (ANC) was associated with 26% higher knowledge, and women who delivered at the health facility were three times more likely to be knowledgeable of ODS [10]. Other factors such as the level of education among women, urban, and occupation were associated with maternal knowledge of ODS [10]. Strengthening ANC and multi-sectoral involvement by ensuring adequate education of the ODS may increase early recognition of complications and timely healthcare-seeking behaviors, thereby significantly decreasing maternal mortality [11,12]. Unfortunately, other factors are unknown in many other SSA countries, which hinders the interventions towards preventing maternal mortality.

In Rwanda, 203 women die for every 100,000 live births [13]. According to the most recent Rwanda Demographic Health Survey (2019–2020), the most common maternal mortality causes are delays in seeking care [11–13]. The value of women's awareness and early recognition of ODS cannot be overstated. Evidence shows that women with better knowledge of ODS have improved healthcare-seeking behaviors than their counterparts with limited knowledge [3,9,14]. Yet, the maternal knowledge of ODS in Rwanda is unknown, limiting any interventions to decrease maternal mortality. Accordingly, the current study aimed to assess pregnant women's knowledge of ODS during pregnancy, labor and delivery, and the postpartum periods and associated factors.

### Objectives

To assess the maternal knowledge of ODS, and associated factors during pregnancy delivery and post-partum period among pregnant women attending ANC services in Kigali, Rwanda.

## Methods

### Ethics statement

Approval was granted from the Institutional Review Board of the College of Medicine and Health Sciences at the University of Rwanda (No317/CMHS/IRB/2018).

And then the approval 2 was granted by Centre Hospitalier Universitaire de Kigali (CHUK) Research and Ethics Committee (EC/CHUK/640/2018).

Participation in the study was voluntary. Pregnant mothers received an explanation of the study's intent and what their participation would entail. Formal written consent was obtained from pregnant women who agreed to participate before the interview. The collected data was

kept confidential on a password-protected laptop only accessible to the research team before the de-identification.

## Study design

We used a cross-sectional study design to assess women's ODS knowledge. To ensure the fullness of the report, we used the STROBE (Strengthening the Reporting of Observational Studies in Epidemiology) checklist [15].

## Study setting

The current study was conducted in the University Teaching Hospital of Kigali (CHUK), Muhima District Hospital, and five community health centers in Nyarugenge District (Muhima, Bilyogo, Rugarama, Kabusunzu, and Rwampara).

## Recruitment and sampling

The recruitment took place between September and December 2018. Conveniently, we sampled women during their ANC appointments at the study sites. Pregnant women of any gestational age, 16 years and older, receiving ANC at the study sites during the data collection period were eligible to participate. The sample size was determined using a single population proportion formula [16], referring to the prevalence of knowledge of at least three ODS of 45.9%, with a 95% confidence level and a 5% margin of error reported by a study conducted in Ethiopia [16].

## Instrument, pilot testing, and data collection

We adapted the study questionnaire developed and initially tested in Ethiopia [17]. The instrument was a 20 minutes questionnaire consisting of 23 items about maternal demographics, obstetrical history, time to the health facility, source of ODS information, and knowledge of ODS separately during pregnancy, labor and delivery, and immediate postpartum periods. All questions were multiple choice. To ensure adapt the instrument to the local context, we removed the item regarding ethnicity and modified the question about religion. Two bilingual speakers of English and Kinyarwanda translated the questionnaire from English to Kinyarwanda. We tested the tool on 30 participants at Muhima health center and in Muhima District Hospital, and there was no need for changes in the instrument. We added the data from the 30 participants of pilot testing to the overall data during analysis.

The investigators (EU, UP, AN, NU) and one trained research assistant (BJ) did the data collection. Each research participant was interviewed alone, and the data collector read questions to the mother and wrote responses on her behalf.

## Data management and analysis

The outcome of interest was maternal knowledge of ODS during pregnancy, labor and delivery, and postpartum. In each category, a mother was classified as knowledgeable if she correctly mentioned at least three ODS from a pool of correct and incorrect danger signs.

Sociodemographic characteristics, including age, level of education, and income, are some of the powerful predictors of health literacy, according to many empirically tested health literacy frameworks [18]. We included them in our assessment of factors associated with the knowledge of ODS. We added predictors specific to the knowledge of ODS, namely, the number of prenatal care visits during the current pregnancy, gravidity, employment status, duration of travel to the health facility, and religion.

We used Stata/BE 17.0 to analyze data. We used descriptive statistical analysis and reported counts and percentages to describe the sample and the knowledge of ODS. We used multiple logistic regression models to assess the factors of ODS knowledge and reported odds ratio and 95% Confidence intervals. The multiple logistic regression model in each category of ODS knowledge was adjusted for all other sociodemographic and pregnancy history factors. The associations with a p-value equal to or below 0.05 were considered statistically significant.

## Results

A total of 382 pregnant women participated in the study. Participants were mostly aged between 16–35 (88.7%), were married or cohabitant (88.2%), had an equal mix of primary or lower education (49.5%) and secondary education or higher (50.5%), and mainly were Catholic or Protestants (72.0%) by religion (Table 1). Two hundred twenty-three (58.4%) were homemakers with no other type of employment. Most women (46.6%) had been pregnant two

**Table 1. Demographic characteristics of participants (n = 382).**

| Variables | n (%) |
|---|---|
| **Age (years)** | |
| 16–25 | 152 (39.8) |
| 26–35 | 187 (48.9) |
| ≥36 | 43 (11.3) |
| **Marital status** | |
| Married or cohabitant | 337 (88.2) |
| Single, separated, divorced, or widow | 45 (11.8) |
| **Religion** | |
| Muslim | 48 (10.5) |
| Catholic | 137 (35.9) |
| Protestant | 138 (36.1) |
| Adventist | 44 (11.5) |
| Others | 23 (6.0) |
| **Education** | |
| Primary or lower | 189 (49.5) |
| Secondary or higher | 193 (50.5) |
| **Number of Prenatal visits** | |
| 0 | 76 (20.0) |
| 1 | 38 (10.0) |
| 2 | 48 (12.6) |
| 3 | 220 (57.6) |
| **Gravidity** | |
| Gravida 1 | 130 (34.0) |
| Gravida 2 | 178 (46.6) |
| Gravida 3 | 74 (19.4) |
| **Employment** | |
| Housewife | 223 (58.4) |
| Other types of employment | 259 (41.6) |
| **Time to the Health Facility** | |
| <30 Min | 89 (23.4) |
| 15–30 Min | 128 (33.5) |
| >30 Min | 165 (43.2) |

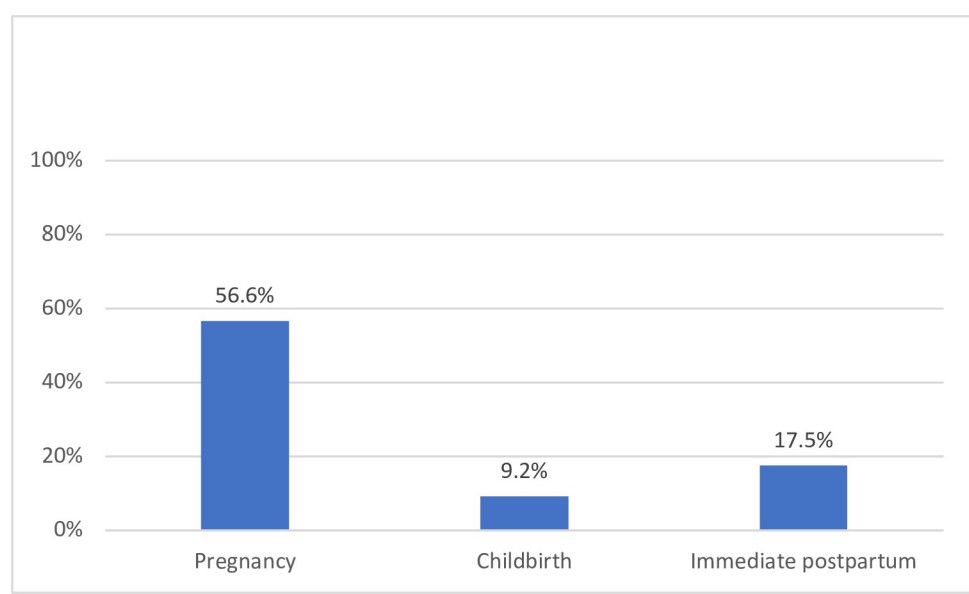

**Fig 1. Overall knowledge of at least three obstetric danger signs.**

times, and 220 (57.6%) attended three ANC visits during the current pregnancy. Most participants (43.2%) walked for more than 30 minutes to the nearest maternal health facility.

Overall, the women's knowledge of ODS during pregnancy, labor and delivery, and postpartum was 216 (56.6%), 35 (9.2%), and 67 (17.5%), respectively (Fig 1). The most recognized ODS during pregnancy were vaginal bleeding, 271 (70.9%), and severe abdominal pain, 196 (51.3%) (Table 2). The least recognized ODS during pregnancy were loss of consciousness, 15 (3.9%); high fever, 19 (5.0%); and convulsions, 26 (6.8%).

During labor and delivery, the most recognizable ODS was vaginal bleeding, 136 (35.6%). Few participants, 45 (11.8%) and 35 (9.2%) recognized that labor lasting longer than 12 hours

**Table 2. Knowledge of ODS during pregnancy, labor and delivery, and immediately postpartum (n = 382).**

| Danger signs | Pregnancy n (%) | Labor & Delivery n (%) | Postpartum n (%) |
|---|---|---|---|
| Vaginal bleeding | 271 (70.9) | 136 (35.6) | 200 (52.4) |
| Severe headache | 80 (20.9) | 15 (3.9) | 32 (8.4) |
| Convulsion | 26 (6.8) | 9 (2.4) | 18 (4.7) |
| High fever | 19 (5.0) | 19 (5.0) | 30 (7.9) |
| Loss of consciousness | 15 (3.9) | 12 (3.1) | 13 (3.4) |
| Blurred vision | 48 (12.6) | n/a | 18 (4.7) |
| Swollen hand/face | 53 (13.9) | n/a | 22 (5.8) |
| Difficulty in breathing | 34 (8.9) | n/a | 19 (5.0) |
| Severe weakness | 110 (28.8) | n/a | 52 (13.6) |
| Severe abdominal pain | 196 (51.3) | n/a | n/a |
| Fetal movement (rapid or slow) | 117 (30.6) | n/a | n/a |
| Water breaks without labor | 103 (27.0) | n/a | n/a |
| Labor lasting >12hr | n/a | 45 (11.8) | n/a |
| Placenta delay > 30 min | n/a | 35 (9.2) | n/a |
| Malodorous vaginal discharge | n/a | n/a | 52 (13.6) |

*Immediate postpartum; n/a—not applicable to trimester on the questionnaire.

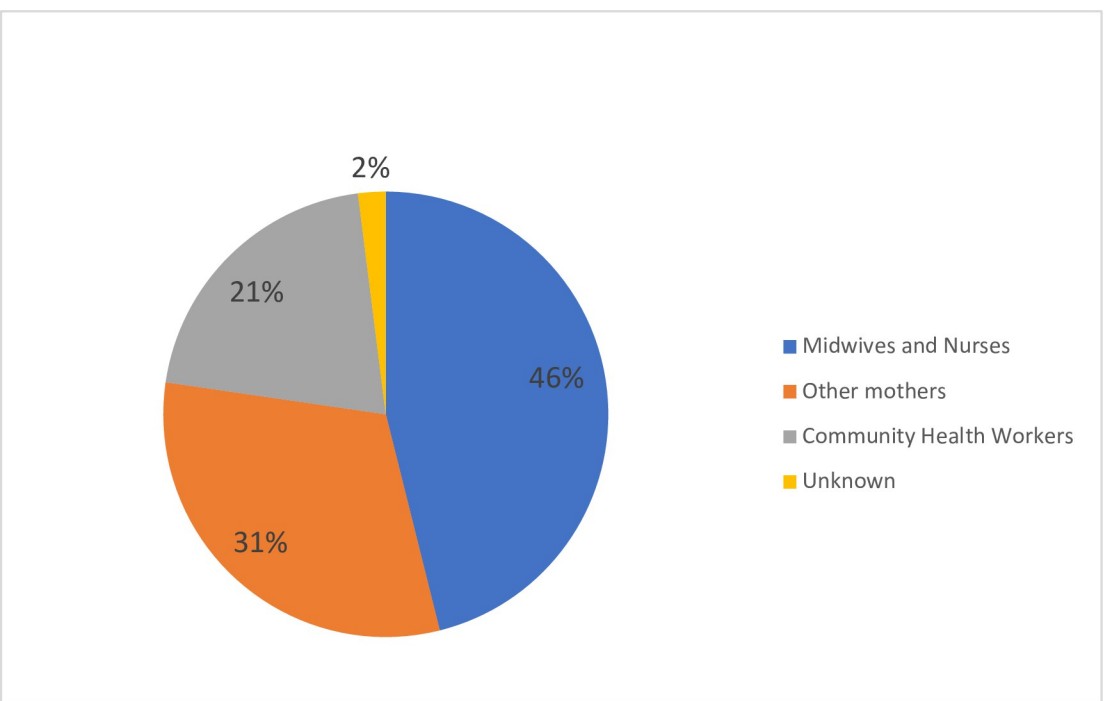

**Fig 2. Source of information about obstetric danger signs (n = 327).**

and a delayed placenta delivery, respectively, are ODS during labor and delivery. For the ODS knowledge during the postpartum period, half of the respondents– 136 (52.4%) were aware that vaginal bleeding postpartum is an ODS. The least recognized OD postpartum were those associated with blurred vision, convulsion, loss of consciousness, and high fever.

Regarding the source of ODS information, 176 (46.1%) got information from nurses and midwives, 119 (31.2%) from fellow mothers, and 79 (20.7%) from community health workers (**Fig 2**).

### Factors associated with ODS knowledge during pregnancy, labor and delivery, and immediate postpartum

The age, marital status, the number of prenatal care visits during the current pregnancy, and the duration of travel to the nearest health facility were associated with the knowledge of ODS during pregnancy (Table 3). Women aged 26 to 35 had 80% higher odds (OR: 1.80; 95% CI: 1.05, 3.06) of ODS knowledge than women aged 16 to 25. Married or cohabitant women had two times higher odds of ODS knowledge than single, separated, divorced, and widows (OR: 0.43; 95% CI: 0.21, 0.89). Attending three ANC visits was associated with 2.6 times higher odds (OR: 2.60; 95% CI: 1.17, 5.66) of ODS knowledge than not attending any ANC visit. Traveling 15 to 30 minutes to the nearest health facility is associated with two times fewer odds (OR: 0.47; 95% CI:0.29, 0.86) of ODS knowledge than traveling for less than 15 minutes. No participant's characteristic was associated with ODS knowledge during labor and delivery.

The knowledge of ODS during the postpartum period was associated with religion, and Muslim women had significantly higher ODS knowledge than Catholics, Protestants, and Adventist women. Muslim women had 2.5 times higher odds (OR 2.4, 95% CI: 1.02, 5.88) of ODS knowledge in immediate postpartum than Catholics.

**Table 3. Factors associated with the knowledge of at least three ODS during pregnancy, labor and delivery, and postpartum.**

| | Pregnancy | Labor and delivery | Postpartum |
|---|---|---|---|
| | Adjusted odds ratio (95% CI) | Adjusted odds ratio (95% CI) | Adjusted odds ratio (95% CI) |
| **Age category** | | | |
| 16–25 | Ref | Ref | |
| 26–35 | 1.80 (1.05, 3.06) * | 2.19 (0.75, 6.37) | 0.96 (0.48, 1.92) |
| ≥36 | 1.75 (0.74, 4.1) | 3.63 (0.91, 14.58) | 0.91 (0.25, 2.65) |
| **Marital status** | | | |
| Married or cohabitant | Ref. | Ref | |
| Single, separated, divorced, or widow | 0.43 (0.21, 0.9) * | 0.88 (0.24, 3.22) | 0.70 (0.25, 2.01) |
| **Education** | | | |
| Primary or lower | Ref. | Ref | |
| Secondary or higher | 1.17 (0.74, 1.84) | 1.82 (0.82, 4.02)) | 1.21 (0.67, 2.17) |
| **Number of Prenatal visits** | | | |
| 0 | Ref. | | |
| 1 | 2.22 (0.92, 5.31) | 0.29 (0.30, 2.95) | 0.40 (0.71, 2.26) |
| 2 | 2.20 (0.91, 5.32) | 1.01 (0.21, 4.97) | 1.46 (0.40, 5.32) |
| 3 | 2.59 (1.17, 5.66) * | 0.95 (0.21, 4.20) | 2.38 (0.76, 7.45) |
| **Gravidity** | | | |
| Gravida 1 | Ref. | Ref | |
| Gravida 2 | 0.92 (0.47, 1.81) | 0.91 (0.25, 3.36) | 1.48 (0.59, 3.70) |
| Gravida 3 | 0.51 (0.22, 1.19) | 1.82 (0.43, 7.73) | 0.78 (0.24, 2.48) |
| **Employment** | | | |
| Housewife | Ref. | Ref | |
| Other types of employment | 0.80 (0.59, 1.26) | 1.15 (0.54, 2.47) | 1.27 (0.71, 2.27) |
| **Time to the Health Facility (minutes)** | | | |
| <15 | Ref. | Ref | |
| 15–30 | 0.47 (0.26, 0.86) * | 0.78 (0.28, 2.17) | 0.74 (0.33, 1.67) |
| >30 | 0.84 (0.47, 1.50) | 1.0 (0.39, 2.59) | 1.24 (0.59, 2.65) |
| **Religion** | | | |
| Catholic | 0.96 (0.45, 2.06) | 0.47 (0.14, 1.55) | 0.41 (0.17, 0.98) * |
| Protestant | 1.04 (0.48, 2.22) | 0.61 (0.19, 1.90) | 0.43 (0.18, 1.02) |
| Adventist | 0.67 (0.27, 1.70) | 0.38 (0.08, 1.85) | 0.28 (0.08, 0.92) * |
| Others | 0.97 (0.32, 2.98) | 0.95 (0.18, 4.89) | 1.34 (0.39, 4.61) |

*p<0.05.

## Discussion

The current study explored maternal knowledge of ODS and associated factors during pregnancy, labor and delivery, and the immediate postpartum periods. Our results indicate that 56.6%, 9.2%, and 17.5% were knowledgeable of ODS during pregnancy, labor and delivery, and immediate postpartum, respectively. The age, marital status, the number of prenatal care visits during the current pregnancy, travel duration to the nearest health facility, and religion were associated with maternal knowledge of ODS.

Our study reported low maternal knowledge of ODS in general, and these findings are consistent with prior similar studies in other settings of SSA. Studies conducted in Ethiopia, Uganda, and South Africa reported that maternal knowledge of at least three ODS was 46.7%, 19%, and 5.2%, respectively [6,8,19]. Bleeding was the most recognizable ODS, consistent with

prior literature. Studies in Uganda, Tanzania, and Ethiopia reported that high women's awareness of vaginal bleeding as an ODS is due to blood-red being symbolized as a danger in African cultures [6–9,11,20–25]. This aspect of the findings is encouraging, given that 27% of all maternal deaths are caused by postpartum hemorrhage [1].

Although infections and hypertensive disorders are among the significant causes of pregnancy-related maternal mortality, the ODS associated with them were the least recognized by women. In Kenya, fever was the third most common ODS mentioned [26] and in Tanzania, headaches during pregnancy were recognized as an ODS by 44% of women [9]. These findings highlight the need to strengthen the women's education of ODS about sepsis and hypertensive disorders during ANC.

The older age was favorable to maternal knowledge of obstetrical danger signs during pregnancy. Usually, older age correlates with a higher number of pregnancies hence higher encounter with the health systems or other opportunities to learn about pregnancy and delivery, such as caregiving to pregnant relatives and friends. Therefore, age is expected to be associated with higher maternal knowledge of ODS. Studies in Ethiopia, Tanzania, and Nigeria reported similar findings of higher age and better knowledge of ODS [27–31]. These findings point to the potential value of structured peer-to-peer learning among pregnant women at the ANC sessions or in the community. This strategy helps older, likely knowledgeable mothers to exchange knowledge and experience with younger and inexperienced women. Peer education is influential in mothers' change or the adoption of certain health behaviors [32] and increased maternal knowledge of ODS effective in Tanzania [33].

Marital status was significantly associated with ODS knowledge, where married or cohabitant mothers are twice as likely to be knowledgeable. A study in Ethiopia reported similar findings [34]. The mechanism by which marital status contributes to maternal knowledge of ODS is not well understood. In the Rwandan culture, becoming pregnant while not married or not living with a partner comes with criticism, stigma, and sometimes rejection by the family, while the pregnancy of married couples is often seen as a blessing. Such cultural dynamics affect women's access to family support and utilization of health services. A study conducted in rural Rwanda reported that single women had three times higher risk for poor utilization of ANC services. Additional studies are needed to explore the association between marital status and women's health literacy on ODS.

In the current study, the higher attendance to ANC was associated with better knowledge of ODS during pregnancy but not during labor and delivery and postpartum. These findings are expected. In Kenya, the ANC attendance was significantly associated with the knowledge of ODS [26]. Similar results have been reported in Ethiopia, Tanzania, and Ethiopia [27–31]. The lack of significant association between ANC attendance and ODS knowledge during labor and delivery and the postpartum period could be a factor in the type of content covered during ANC visits. These findings point to the need for strategies that promote early and sustained attendance to ANC. There is additionally the need to ensure that the content covered during ANC visits includes the maternal understanding of the labor and birth procedure and postpartum expectations, as well as possible complications during these periods.

The accessibility of health services is an essential factor in health literacy and health outcomes [35]. Our study found that longer travel to the nearest maternal health facility was significantly associated with less knowledge of ODS during pregnancy. Even when health services are available, their utilization remains contingent on their accessibility by geographical location and cost of care [36]. Traveling longer distances is often associated with extreme physical exhaustion through walking or high travel costs, and the loss of time. The accessibility of health services partly explains why rural residents are likely to be less health literate than their urban counterparts [35]. These findings highlight the value of decentralizing health services closer to

the community and universal health coverage to alleviate health geographical and cost barriers to health services' accessibility.

We additionally found significant disparities in ODS knowledge by religion during postpartum but not pregnancy and labor and delivery periods. In our study, religion relates to the type of church where people worship. Muslims were significantly different from the rest of the study participants, yet, there is no known explanation for this finding. The science behind religious beliefs and health literacy is still in its infancy. In a study conducted in the United States of America, higher religious beliefs scores were associated with lower health literacy about colorectal cancer [37]. Furthermore, there is a need to understand why this type of association existed only during the postpartum period. Additional studies exploring the role of religion on ODS knowledge are needed.

Despite our call for additional studies on ODS knowledge, the current literature lacks consistency in defining this concept. Some studies define maternal knowledge as the recognition of three ODS, while others raise the threshold to the knowledge of a higher number of ODS. The variability in reporting renders the comparability of findings ambiguous, which calls for a universal instrument for assessing maternal knowledge of ODS.

## Limitations

This study used a cross-sectional design; hence we cannot learn the trend in maternal knowledge of ODS across the trajectory of pregnancy, labor and delivery, and postpartum. Since the study participants were mainly the residents of Kigali–the capital city of Rwanda, the findings may not be generalizable to other regions, especially the rural population. It was an observational study; therefore, we cannot make causal inferences between explanatory variables and ODS knowledge. The comparably low knowledge of ODS during labor and delivery and postpartum than during pregnancy could be explained by the fact that the study participants were pregnant, so familiarity with ODS in the other perinatal periods may have been less critical at the time of data collection.

## Conclusion

This study assessed pregnant women's knowledge of obstetrical danger signs during pregnancy, labor and delivery, and immediate postpartum. Maternal knowledge was found to be low, especially the knowledge of ODS during labor and delivery and the immediate postpartum period. Early in pregnancy, women need to be introduced to what to expect during labor and delivery as well as the postpartum period. This understanding could help them know potential complications and recognize them early and seek support if they occur. Other media, including public service announcements on TV and radios, could be leveraged to sensitize the general knowledge about pregnancy, labor and delivery, and the postpartum period.

Young mothers, women without partners (single and widow), and those traveling long distances to health facilities are at increased risk for low ODS knowledge, and particular interventions, including peer-to-peer support and decentralization of maternal services, are needed. Additional studies are also needed to understand the association between ODS knowledge and marital status and religion.

## Supporting information

**S1 Data. Danger signs.**
(CSV)

## Acknowledgments

We would like to thank the midwives and health administrators at the health facilities for their cooperation with the data collection, Daniel Bogale, the Assistant professor who gave us the structured questionnaire, and Becky White, who helped in structuring the title of this research project.

## Author Contributions

**Conceptualization:** Emmanuel Uwiringiyimana, Albert Nsanzimana, Neophyte Uhawenayo, Pacifique Ufitinema, Patricia J. Moreland.

**Data curation:** Emmanuel Uwiringiyimana, Samuel Byiringiro, Janviere Bayizere.

**Formal analysis:** Emmanuel Uwiringiyimana, Samuel Byiringiro.

**Investigation:** Emmanuel Uwiringiyimana.

**Methodology:** Emmanuel Uwiringiyimana, Emery Manirambona, Samuel Byiringiro, Patricia J. Moreland, Pamela Meharry.

**Project administration:** Emmanuel Uwiringiyimana, Albert Nsanzimana, Neophyte Uhawenayo.

**Resources:** Emmanuel Uwiringiyimana.

**Software:** Emmanuel Uwiringiyimana, Janviere Bayizere.

**Supervision:** Emmanuel Uwiringiyimana, Samuel Byiringiro, Patricia J. Moreland, Diomede Ntasumbumuyange.

**Validation:** Emmanuel Uwiringiyimana.

**Visualization:** Emmanuel Uwiringiyimana, Janviere Bayizere.

**Writing – original draft:** Emmanuel Uwiringiyimana, Samuel Byiringiro, Albert Nsanzimana, Neophyte Uhawenayo, Pacifique Ufitinema, Pamela Meharry.

**Writing – review & editing:** Emmanuel Uwiringiyimana, Emery Manirambona, Samuel Byiringiro, Pacifique Ufitinema, Patricia J. Moreland, Pamela Meharry, Diomede Ntasumbumuyange.

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
