## [Decision Letter · Decision Letter 0]

23 Nov 2021

PGPH-D-21-00713

Pregnant women's knowledge of obstetrical danger signs: A cross-sectional survey in Kigali, Rwanda

Dear Dr. Uwiringiyimana,

Thank you for submitting your manuscript to PLOS Global Public Health. After careful consideration, we feel that it has merit but does not fully meet PLOS Global Public Health’s publication criteria as it currently stands. Therefore, we invite you to submit a revised version of the manuscript that addresses the points raised during the review process.

We look forward to receiving your revised manuscript.

Kind regards,

Zohra S. Lassi, PhD

Academic Editor

Journal Requirements:

1. Please provide additional details regarding participant consent. In the ethics statement in the Methods and online submission information, please ensure that you have specified whether consent was informed.

2. Please provide separate figure files in .tif or .eps format only, and remove any figures embedded in your manuscript file.  If you are using LaTeX, you do not need to remove embedded figures.

3. Please update the completed 'Competing Interests' statement, including any COIs declared by your co-authors. If you have no competing interests to declare, please state "The authors have declared that no competing interests exist". Otherwise please declare all competing interests beginning with the statement "I have read the journal's policy and the authors of this manuscript have the following competing interests:"

4. In the online submission form, you indicated that "The collected data was kept confidential, on a Password Protected laptop only accessible to the research team prior to the deidentification" All PLOS journals now require all data underlying the findings described in their manuscript to be freely available to other researchers, either 1. In a public repository, 2. Within the manuscript itself, or 3. Uploaded as supplementary information.

Additional Editor Comments (if provided):

Dear Authors,

This paper is not providing any new information and therefore suggest can be taken and published in region specific or local journals.
---

## [Decision Letter · Decision Letter 1]

5 Jul 2022

PGPH-D-21-00713R1

Pregnant women's knowledge of obstetrical danger signs: A cross-sectional survey in Kigali, Rwanda

Dear Dr. Uwiringiyimana,

Thank you for submitting your manuscript to PLOS Global Public Health. After careful consideration, we feel that it has merit but does not fully meet PLOS Global Public Health’s publication criteria as it currently stands. Therefore, we invite you to submit a revised version of the manuscript that addresses the points raised during the review process.

We look forward to receiving your revised manuscript.

Kind regards,

Zohra S. Lassi, PhD

Academic Editor

Journal Requirements:

Additional Editor Comments (if provided):

as per the reviewer the authors have not addressed the comments and I would like to give you one more chance to address before we formally reject as per the decision.

Reviewers' comments:

Reviewer's Responses to Questions

**Comments to the Author**

1. If the authors have adequately addressed your comments raised in a previous round of review and you feel that this manuscript is now acceptable for publication, you may indicate that here to bypass the “Comments to the Author” section, enter your conflict of interest statement in the “Confidential to Editor” section, and submit your "Accept" recommendation.

Reviewer #1: (No Response)

2. Does this manuscript meet PLOS Global Public Health’s publication criteria? Is the manuscript technically sound, and do the data support the conclusions? The manuscript must describe methodologically and ethically rigorous research with conclusions that are appropriately drawn based on the data presented.

Reviewer #1: No

3. Has the statistical analysis been performed appropriately and rigorously?

Reviewer #1: No

4. Have the authors made all data underlying the findings in their manuscript fully available (please refer to the Data Availability Statement at the start of the manuscript PDF file)?

Reviewer #1: No

5. Is the manuscript presented in an intelligible fashion and written in standard English?

Reviewer #1: No

6. Review Comments to the Author

Reviewer #1: It is a bit unclear to me why this study is relevant for an international readership given that similar studies were conducted already in three other countries. The study is also limited to the capital city, in the sample size, and using descriptive methods.

The underlying causal link between knowing a danger sign and maternal death needs to be strengthened. It is possible that a woman cannot name a danger sign but still acts when she is bleeding or in severe pain. Thus I found that important to emphasise that if they are unsure about these signs they might delay action. Also please refer here to the fact that seeking healthcare is costly that is why they need to be more sure when to act.

Stating that maternal death is high worldwide is not true. It is quite low in high-income settings which further emphasises that this would be mostly preventable.

The discussion needs to be better structured. Also, you draw conclusions that I believe are not supported by your findings such as “Encouraging the involvement of husbands in all perinatal visits at health centers and hospitals has the potential to improve outcomes, as well as encouraging legal marriage”. You did not really investigate this issue. =This being said I agree with you but it's not your conclusion).

The sample size is rather small and does not allow multivariate analysis. Sample size should be mentioned in the abstract.

You state that the inclusion criterion was ‘aged 18 years and older’ but you 11 participants under 18 in Table 1.

It is unclear to me how you asked about danger signs. Was it an open-ended question or a closed question? Maybe women cannot name blurred vision as a problem by themselves but when you asked them whether it is a problem they will name it as a problem. Please specify how you exactly asked about these and possible make the questionnaire available.

There are some typos

7. PLOS authors have the option to publish the peer review history of their article (what does this mean?). If published, this will include your full peer review and any attached files.

**Do you want your identity to be public for this peer review?** For information about this choice, including consent withdrawal, please see our Privacy Policy.

Reviewer #1: No

---

## [Editor Report · Decision Letter 2]

19 Oct 2022

Pregnant women's knowledge of obstetrical danger signs: A cross-sectional survey in Kigali, Rwanda

PGPH-D-21-00713R2

Dear Mr Uwiringiyimana,

We are pleased to inform you that your manuscript 'Pregnant women's knowledge of obstetrical danger signs: A cross-sectional survey in Kigali, Rwanda' has been provisionally accepted for publication in PLOS Global Public Health.

Best regards,

Zohra S. Lassi, PhD

Academic Editor
